# To Obtain Informed Consent or Not to Obtain Informed Consent? Drones for Health Programs in the Grey Zone between Research and Public Health

Vyshnave Jeyabalan [1,2,*], Lorie Donelle [3], Patrick Meier [4] and Elysée Nouvet [2]

1 Department of Health Information Science, Faculty of Information and Media Studies, Western University, London, ON N6A 3K7, Canada
2 School of Health Studies, Western University, London, ON N6A 3K7, Canada
3 Arthur Labatt Family School of Nursing, Western University, London, ON N6A 3K7, Canada
4 WeRobotics, 1204 Geneva, Switzerland
* Correspondence: vjeyabal@uwo.ca

**Abstract:** Drones are increasingly being introduced to support healthcare delivery around the world. Most Drones for Health projects are currently in the pilot phase, where frontline staff are testing the feasibility of implementing drones into their healthcare system. Many of these projects are happening in remote localities where populations have been historically under-served within national healthcare systems. Currently, there exists limited drone-specific guidance on best practices for engaging individuals in decision-making about drone use in their communities. Towards supporting the development of such guidance, this paper focuses on the issue of obtaining community and individual consent for implementing Drones for Health projects. This paper is based on original qualitative research involving semi-structured interviews (*N* = 16) with program managers and implementation staff hired to work on health-related projects using drone technologies. In this paper, we introduce a scenario described by one participant to highlight the ethical and practical challenges associated with the implementation and use of drones for health-related purposes. We explore the ethical and practical complexities of obtaining informed consent from individuals who reside in communities where Drones for Health projects are implemented.

**Keywords:** drones; healthcare delivery; informed consent; collective consent; ethics





## 1. Introduction

There is no doubt that the ideal of voluntary and informed individual consent in health research is fraught: what constitutes 'voluntary' participation in contexts where health research may provide hope or actual promise of healthcare in the context of limited options? Why consider individual consent across all societies where individual decision-making may not be the norm? What constitutes 'informed' decision-making? How much information must be provided and understood for such decisions to be informed? Despite these questions being a perennial source of debate in the bioethics community, informed and voluntary consent constitutes a core principle for the conduct of all research involving human subjects [1–3]. This is with good reason. The requirement of informed and voluntary research participation emerged in the aftermath of several ethical failures in science, including Nazi experiments during World War II on prisoners and the U.S.-based Tuskegee experiments on African American "volunteers" left to suffer treatable syphilis in the name of scientific observation until the 1970s [4]. Today, no human research can proceed without informed consent: this requires providing potential participants with clear information about what research participation involves, giving participants enough time to consider participation, not coercing or influencing their choice, and highlighting the choice to refuse participation [1]. This paper considers whether similar requirements should be expected in the context of drones for health projects.

Unlike research-based initiatives, voluntary and informed consent is not a requirement for all public health initiatives. Obtaining informed consent from individuals for public health initiatives may not be feasible or appropriate [5–7]. Providing individuals with the opportunity to consent to public health interventions may undermine the benefits for others in the society [5,8]. For example, it may be that one individual's interests do not align with community needs, and the benefit to the community outweighs the precedence given to individual autonomy. In such instances, informed consent is not appropriate [5,6]. Informed consent may not be appropriate in situations where consent practices create privacy risks for individuals [7]. In Ontario, Canada, if a public health initiative poses no more than minimal risk and does not involve therapeutic, clinical, or diagnostic interventions, then informed consent is not required [7]. In some public health scenarios, consent may be appropriate and necessary for some elements of an intervention but not for others. For example, routine surveillance activities that indirectly collect patient information do not require informed consent; however, non-surveillance activities may require consent [6,7]. De-identified personal information used for public health purposes does not raise autonomy issues because the information is not linked to an individual's identity, so there is no right-based justification for an individual to control that information [5]. In contrast, exemptions can be made to sharing identifiable personal information in cases such as contagious disease tracing [5]. It can be seen that even within the realm of public health initiatives, when to and when not to obtain an individual's informed consent is complex and context-specific.

A novel public health initiative includes introducing drones to support healthcare delivery. Around the world, drones are being used for various healthcare and public health purposes, including delivering medical supplies to restock pharmacies and local clinics, delivering biological samples to expedite laboratory testing and treatment, making public health announcements to remind individuals to maintain the recommended six feet distance during the COVID-19 pandemic, and monitoring the environment to assess and manage wildfires and floods [9–17]. Many of these Drones for Health projects are happening in remote localities, where populations have been historically under-served within national healthcare systems [18]. However, most Drones for Health projects remain in the pilot phase, with drone teams testing the feasibility of implementing this technology in specific healthcare systems or contexts [18].

Currently, there are limited guidelines that describe the use of drones [19,20]. The Humanitarian UAV Code of Conduct describes data protection, community engagement, effective partnerships, and conflict sensitivity in the context of humanitarian drone use [19]. However, this guideline does not include healthcare-specific considerations to facilitate responsible and ethical drone use specifically for these healthcare projects [19]. Instead, these guidelines make general recommendations for the use of drones for humanitarian deployment [19]. In 2020, WeRobotics did develop a training program entitled "Community Engagement for Social Goods Projects with Drones", in which interested individuals could pay for and enrol [20]. This course describes the principles of community engagement, approaches to community engagement, and overcoming challenges during community engagement. However, these guidelines and training programs are not specific to the use of drones in the context of healthcare, and it is unclear how widely they are consulted and used by Drones for Health implementers.

Despite the above guidance, there has been no explicit discussion to the best of our knowledge on whether Drones for Health programs embedded in public health programs should seek consent from individuals or communities where these programs are being introduced. This is further complicated when there is ambiguity around whether these Drones for Health projects fall under public health initiatives or research.

Towards initiating a broader discussion of consent expectations in the context of Drones for Health projects, this paper describes several practical, ethical, and sociocultural considerations that may emerge when drones are introduced into remote communities. The point of departure for this exploration is one scenario shared with the first author during a global qualitative research study on ethical and practical complexities within Drones for

Health projects [18]. Overall findings from that study have been published previously [18]. The scenario detailed and analyzed in the present article, however, is sufficiently complex to merit its own analysis. Based on one study participant's description of events, this is a scenario wherein a drone used for a health project was introduced with limited community consultation. The individual who shared this account felt that they walked away uncertain that their team's approach had been the right one. It is difficult, if not impossible, to discuss when, how, under what conditions, and with what rationales individuals responsible for introducing drones for health projects to communities may or may not engage in particular consent-seeking practices with affected community members in the absence of concrete and detailed examples. A closer look at the circumstances this individual and their team were navigating, and retrospective consideration of available options, provides rich grounds for clarifying ethical and practical complexities related to current consent practices in the realm of Drones for Health projects. Following a description and analysis of this scenario, we conclude the article with recommendations that may serve to inform those engaged in Drones for Health projects moving forward.

## 2. Materials and Methods

This article draws on results from a qualitative perceptions study involving semi-structured in-depth interviews with individuals (*N* = 16) from nine countries working on the frontline of Drones for Health programs. The goal of the original qualitative study was to better understand the context-specific concerns, challenges, and complexities of using drones for healthcare purposes. Qualitative research provides insight into experiences, relationships within, and the functioning of healthcare initiatives [21,22]. This multi-sited comparative perception of healthcare study replicates an approach commonly used in program design and quality improvement in the healthcare and humanitarian aid sector [22]. Perception studies are often used in healthcare to understand how frontline health staff, patients, policy-makers, and communities view healthcare initiatives, providing insight into the satisfaction, perceived advantages and disadvantages, and perceived importance of health programs [23–25]. A more detailed description of the research methods used in this study can be found in the study performed by Jeyabalan [18].

### 2.1. Recruitment and Sampling Strategy

Recruitment involved purposive and snowball methods. To participate in this study, participants had to: (1) work in a role that involved responsibility for introducing and implementing Drones for Health programs; (2) be willing and able to participate in a one-hour individual virtual interview. Snowball sampling was also used, as participants were asked if they had colleagues working in similar capacities either in that country or in another country context that might be receptive to an invitation to participate. Participants include those in leadership positions, advisors, technical staff, and researchers.

### 2.2. Data Collection and Data Analysis

Interviews were conducted between June 2019 and February 2020 by conventional phone or Skype by two members of the study team (V.J. and E.N.) in English, Spanish or Nepali. Interviews lasted between 20 and 140 min, with an average of 78 min. Interviews were transcribed verbatim, and where necessary, translated into English. Data analysis was conducted on NVivo 12 (QSR) [26]. Directed and interpretive thematic analysis was used to analyze interviews.

### 2.3. Ethics

This study received approval from the Western University's Research Ethics Board (protocol approval #113823).

## 3. Results

### 3.1. A Study Participant's Experience

A team working on Drones for Health projects went into several remote communities to perform a mapping project to help understand the hazards, health risks, and safety concerns that are caused by the flooding of a nearby forest swamp. Normally, the community leaders tell the community in advance of the drone team's arrival during the community's weekly service. The team's community engagement strategy involved two steps for this context. First, they would present themselves to the elder of the village or the landowner considered responsible for the village and seek their permission to be in the village by participating in a traditional ceremony. Then, the team would engage the community by hosting a gathering or having a drone demonstration to describe the project to the local community members and field questions. In one situation, there had been a recent death in the village, so the village was empty the day the team went into the community to execute the drone project. The team adjusted their plan: they participated in the usual small ceremony and received approval from the village leadership to fly the drone and take pictures of the village for the mapping project. Since most of the villagers were away, the team did not hold a gathering nor conduct a drone demonstration—they just did their job and left.

### 3.2. Practical Challenge: Contacting Local Community Members

The participant in this scenario and their team normally engage in community consultations prior to launching a drone project. These normal engagement activities include, in their account, holding gatherings and drone demonstrations to inform the local community members about the project. On the day from which the above scenario is drawn, the team could not hold a consultation as the community members were away for a funeral.

Other participants (*n* = 11) in this investigation also reported on the challenge of not being able to engage all the community members in these community engagement activities [18]. Conditions contributing to this lack of community engagement included: difficulty engaging all community members due to the community's large population size; community members, especially men, being away at work during the community engagement activities; a reported lack of interest amongst community members in the project due potentially to a sense it was not relevant to their health needs; weather; and insufficient advertising of the engagement activities. One resolution to reaching all stakeholders was to rely on community leaders, women, and children to inform those who could not attend the community engagement activities. However, a lack of first-hand contact with community members may lead to misinformation about projects. At the same time, working through existing channels of spreading information in a community aligns with the principles of community engagement [27,28].

Ethically, not engaging in the usual consultation processes prevented the team in the above-mentioned scenario from being able to provide first-hand information about the project to all the local community members so that they can understand and consent to the project. This raises the question of what constitutes 'good' community consultation and an appropriate consent process? Is obtaining consent from leaders sufficient? Is it appropriate to obtain consent from community gatekeepers, such as locally elected officials, health workers, or elders—those who hold real or symbolic power in the community? Is obtaining approval for a project from, e.g., 30% of local members of a community sufficient to declare a project as one that engaged in and obtained consent from that community? What information do drone teams have to share with the community? How do drone teams ensure that the community understands the project and is capable of providing consent? Furthermore, some might ask whether consent is needed at all? This was, after all, a public health intervention. What is at stake ethically and practically when consent is not obtained in such public health interventions?

### 3.3. Ethical Issue: Lack of Consent

3.3.1. Is Consent Needed at All?

Participant 01, who was involved in the above-mentioned scenario, was asked what they would do differently in the next community and said:

"I mentioned earlier, there was a death in that village and that headman allowed us to work in that community, but there was very few people in the village at the time. So, if I was a resident of that village, I would have had an issue with that because we weren't there and then you were taking pictures of our property. But I am not really sure how these village elders are communicating with their village citizens at the end of the day. I mean the level of respect that the communities have for each other is admirable and when such things happen, then they often do not speak up. But I have heard instances where if they are not comfortable with anything that they do, they raise their concerns."

Due to the death in the village, the participant and team's schedule was pushed back, so they were limited in time and had to go into the community that day to conduct the mapping project. However, it is evident from the quote above that the participant found it problematic to go into the communities and take pictures of local community members' private property without talking to them first. What can and should be expected in such a situation?

It is important to note that similar Drones for Health projects are defined differently within and across countries. Some of the individuals interviewed worked on Drones for Health projects that had received national and sometimes other institutional ethics approval as research projects. Five of the nine Drones for Health projects explored in the larger research project sought research ethics approval, and the other four projects did not. Though some of these Drones for Health projects did not seek research ethics approval and were categorized as public health initiatives, participants still described them as experimental research projects:

" ... , let's make sure that whatever we're transporting actually gets to the destination because at the end of the day we're testing technology, but we needed to make sure that everything was running smoothly ... "—Participant 04

"For example, one of the reasons, because it was created, is for example, to state that it exists a project with drones that it's being developed in the [national] forest and what is missing is to generate evidence so the decision-maker or the State can say, I can incorporate this, but evidence is needed because is more like a bet, right?"—Participant 13

When speaking to individuals who implemented Drones for Health, there emerged definite ambiguity among some of those responsible for running these projects as to whether their work with drone-based innovations constituted research. In several interviews ($n = 6$), including both those with research ethics approval and those without, the terms "study", "research" and "program" were used interchangeably. Most participants framed these pilot projects as a proof-of-concept to generate knowledge about the feasibility of scaling and integrating the project as they were trying to collect information and generate evidence on the workability of drone technology for health purposes in specific contexts and identify whether drone delivery impacts the integrity of medical supplies ($n = 15$). All projects could be seen to impact the day-to-day life of target communities.

The varying consent procedures now provide an opportunity for drone teams to consider the processes that constitute ethical research requirements, which may also reflect 'best practices' for optimal implementation of drone use for health projects. To identify whether an individual's informed consent is required, organizations involved in implementing these Drones for Health projects need to clearly define them either as research projects or public health initiatives. Generally, public health initiatives involve applying proven methods to protect and improve the health of a community, whereas research

involves testing new treatments or strategies that are not known to be efficacious in generating new knowledge [29,30]. Research ethics require researchers to obtain informed consent by (1) providing participants with all the relevant information about the research; (2) refraining from using deception, undue influence, or coercion; (3) ensuring that participants have enough time to consider whether they want to participate and; (4) obtaining signed individual informed consent, unless an exception is warranted and vetted through the research committee [1]. The guideline above may help identify how to respectfully implement Drones for Health projects. It will also help project leaders determine which protocols (i.e., Public Health Initiative vs. Public Health Research) they need to abide by and what processes (i.e., consent for participation) they would be accountable for with the local community members.

However, it is important to note that even if Drones for Health projects are considered public health initiatives, the type of data collected during these projects may impact the need for consent [5,7]. For example, individuals' informed consent is required if Drones for Health projects include non-routine surveillance activities, directly collect information from individuals, impede personal autonomy, adversely affect individual's welfare, or involve therapeutic, clinical, or diagnostic intervention [5,7]. In the presented scenario, the team used the drone to take pictures of residents' private properties—this means that the team may have potentially executed a project without the appropriate approval. Residents did not receive an opportunity to understand how the data that were collected were to be used, who would have access to the data, how the data would be protected, and how it would impact them. This further exacerbates the risks the team posed to community members due to the lack of community consultation.

Determining which public health initiatives do and do not necessitate an individual's informed consent is complex and context-specific [5,7]. This calls for a need to better categorize Drones for Health projects based on their use case to determine the appropriate informed consent process. Until such clarification occurs, it may remain difficult to determine whether individual consent is required for certain Drones for Health projects. Regardless of whether it is a research project or public health initiative, best practices dictate that affected populations should be made aware of Drones for Health projects in their region through community engagement or consultation activities [5,7].

### 3.3.2. If Consent Is Needed for Drones for Health Research, and Is Approval from the Elder in a Community Sufficient?

Three study participants from various Drones for Health projects described challenges with contacting all the local community members because of the community's population size, community member's lack of interest in the project, individuals being busy during community engagement activities, and community members being unaware of community engagement activities (*n* = 3). In the described case scenario, the team substituted the approval from the elder for consent from the individuals living in the community. This raises the question: if consent for research is needed, is it acceptable to obtain consent from the community leader and not from individual affected community members as well?

Personal autonomy is emphasized in Western, industrialized countries. This extends to normative expectations that, in research, no individual can be made to participate in research against their will. Moreover, all research participants should be informed of what is involved in participation and provided with the opportunity to refuse participation. Consent by a community leader may be appropriate in certain settings, but this generally is not accepted as a substitute for individual consent [1,2,31,32]. At the same time, the relationship between collective/community leaders and individual consent is rather underdeveloped. It is important to note that the ethical principles of respect for autonomy, non-malfeasance, beneficence, and justice are shared across diverse cultures; however, the application of these principles requires context-specific consideration of the local values and philosophies [33]. Does acting on community leaders' approval of Drones for Health projects in the absence of individual consent constitute unethical practice? Or is an assump-

tion of needing individual consent for the trialing of new drone technologies for health potentially disrespectful of cultural norms that favor collective decision-making? It is far from clear but worth considering arguments for and against relying on community leaders' approval.

In many non-Western cultures and communities, it is normative for family members or community leaders to play a significant role in individuals' decision-making [32,34]. In certain communities, collective consent is endorsed and preferred to individual consent [31,33]. Collective consent is the process of obtaining consent from community representatives who have the responsibility to act in the best interests of the community members through processes such as community consultation [31,33]. For example, the Indigenous communities use collective decision-making to complement individual consent [31]. Communities are requiring researchers to obtain collective consent so that they can exercise their right to determine which research projects are being undertaken on their people and lands [33,35]. Collective consent attempts to decolonize current research ethics processes by allowing communities to have control over the research projects being conducted [35]. Collective consent also provides communities the opportunity to assess the implications the research project has on the wider community and partake in the whole decision-making process of the project [35,36]. It allows communities to develop meaningful relationships with researchers and ensures that researchers are utilizing culturally appropriate ethical parameters for the project [33]. Many national research guidelines, such as those from Canada, Australia, and New Zealand, highlight the importance of consultation and the need for both community and individual-level consent for executing research initiatives when working with Indigenous communities [1,31,33,37].

The idea of collective consent is relevant in many collectivistic societies. In fact, the principle of respect for autonomy may be perceived as culturally insensitive in these collectivistic societies that place importance on the benefit of the group as a whole [38,39]. In many of these Drones for Health projects ($n = 7$), drone teams have approached traditional or elected leaders to obtain approval to conduct the project prior to talking to community members: 'The traditional king in the village has the main role in the communities for [approving] any intervention in the village.' (Participant 06). Study participants have identified that to appropriately respect local culture and to earn local community members' trust they need to obtain approval from a community representative ($n = 6$). A survey indicated that one-third of researchers from the United States performing international research sought approval from a village leader to conduct their research [40]. Researchers who conducted a study in Kenya observed that in order to gain permission to conduct a project, they need to inform the village chief and elders about the study [41]. In these settings, investigators are expected to accommodate local customs and cultures and seek approval from the appropriate community leader to implement the Drones for Health projects [1,32,34]. However, there are challenges to obtaining collective consent when community consultation is not performed appropriately. This often occurs when teams have limited knowledge of the local culture, are disrespectful, are dishonest, do not speak to the appropriate representatives for community consultation, and are limited in time [33]. Furthermore, the appropriateness of community consultation varies on the local sociocultural and political conditions, and thus, drone teams need to consider the local context to determine the appropriate consent practices.

Different consent processes have been utilized within the same country and different community settings ($n = 1$). For example, in rural villages, the drone teams obtained collective permission to perform the project, and in urban settings, they obtained consent from each household ($n = 1$). Additionally, some project teams obtained consent from the traditional leader of the community to use private property for the project ($n = 3$), whereas other teams obtained consent from the actual landowners ($n = 6$). This suggests that a broad range of practices are used to obtain consent for the Drones for Health projects based on the community setting. When informed consent practices are inconsistent, it means that some individuals are able to voluntarily participate in the health projects while others

are not, thus undermining an individual's autonomy and creating uncertainty for project teams. This is an important consideration for specific projects, such as mapping projects where large areas of land are photographed, and consent is obtained from the elderly of the village, but it infringes the rights and privacy of multiple landowners.

A participant described that in areas such as settlements with no formal hierarchy, they had to identify a representative of the community to obtain approval. The participant mentioned that they received approval to undertake the project from an individual that some of the community members identified as a representative:

> "I honestly do not know because if it was a structured village then that person would have been chief or somebody who was knowledgeable, but in this settlement, I have no idea why they pointed at him. Maybe it's because he has more rich educational background compared to the rest or maybe he's just somebody who people—everybody respects."—Participant 01

This raises the question of how the team can be confident that everyone in the community considers this individual to be the representative of their community. Identifying appropriate representatives for the community is challenging, especially in communities that do not have a clear representative [5,7,42]. Drones for Health project leaders need to consider how representatives are chosen, how many are selected, and the scope of their power [43]. The Humanitarian UAV Code of Conduct also highlighted that some communities may be marginalized and not represented, so drone teams must understand the local dynamics to identify appropriate community representatives [19]. It is also important that the community representative adequately understands the project, any associated risks, and whose interests are served by these public health projects so that they can make an informed decision about participating in the project [32]. The consequences of obtaining approval from a single non-representative individual may result in decisions that are not in the best interests of community members and discriminate against those who are the most vulnerable and have the greatest needs [19]. These consequences highlight the importance of community consultation instead of just relying on obtaining community gatekeepers' approval.

Finally, substituting several individual community members' consent with the consent from a community gatekeeper's approval can cause issues due to the power relationship between community members and community gatekeepers. It can inadvertently reinforce hierarchies and power imbalances that already exist in these communities [44,45]. Individuals may not be able to disagree with the community gatekeepers and may feel coerced into participating in the project, impacting their ability to freely provide consent to participate in this project [44].

In the scenario described at the outset of this section, the team member describing the scenario walked away, uncertain of whether reliance on the community leader's approval was appropriate. This uncertainty arises within a wider landscape of uncertainty: about the need for consent in the first place, but also in a context where consent processes are inconsistent. This suggests that teams may benefit from opportunities for in-depth discussion and troubleshooting challenges related to current informed consent practices. This could potentially help them identify the most appropriate protocol to obtain consent for Drones for Health projects. If informed consent (for research-based projects) is necessary, teams need to contact individuals who will be impacted by the Drones for Health projects. These individuals will include leaders but also ordinary residents, families, patients, and local community members. Figure 1 describes some of the considerations and complexities that have been raised.

| | |
|---|---|
| Does this project require informed consent? | Clarify the nature of the drone for health project.<br><br>Is this a research or a public health project? If clearly research, get the approval needed from relevant research ethics authorities + population(s) involved. When a project more clearly a public heath initiative, there may be cases where it is permissible for a public health initiative to proceed without the approval of its protocol by a research ethics committee or informed consent from those affected. In such scenarios, it is still important to seek permission and approval of the intervention from affected communities. |
| What are relevant cultural preferences and norms for launching such projects? | Identify context-specific consent-seeking norms for projects with community-level impacts. If there is a preference and norm of "outsiders" seeking consent for projects first with a community leader, learn what and who that procedure ideally involves in dialogue with individuals in the community to be engaged. Failing to adhere to cultural and context-specific expectations for sharing information about and seeking approvals for a drones for health project may result in eroded trust or otherwise compromise the project. |
| Who needs to engaged? | Consult with trusted sources (e.g. local researchers or community engagement experts) to learn about power dynamics and the ways in which norms of decision-making for community-level projects might not take into account the perspective of less powerful members of a community.. Is community-level approval only for your project appropriate? Does the project infringe on individual affected population members' land or rights? |
| What practical factors could limit engagement? | Plan for ways to mitigate practical challenges that, if left unaddressed, could seriously compromise the extent and quality of consultations with individuals that are impacted by drones for health projects. Practical challenges will vary from context to context. These may include, for example, weather conditions, limited time to seeking approval from individuals and communities, or community members being unavailable to participate due to work schedule. Just as these practical considerations will vary from context to context, so will the most culturally acceptable and realistic strategies for addressing these vary from project context to project context. |

**Figure 1.** Drones for Health informed consent considerations.

## 4. Discussion and Conclusions

In the scenario at the center of this paper, the drone team was unable to proceed with their normal consent process due to a funeral, which, as a result, meant that the team ran into the challenge of not being able to contact the local community members. The Drones for Health projects are defined by participants as either a research project or public health initiative—how these projects are categorized impacts the consent practices. One of the key requirements for obtaining informed consent in a research initiative is to provide individuals with all the information about the project so that participants can assess whether they want to participate in the project.

In the scenario above, the project team worked with the traditional leaders to obtain collective consent to move ahead with the drone project. It was a practical and culturally acceptable solution to a practical problem. Moreover, it could be argued that the failure

to respect the authority of traditional or elected leaders could represent a sort of ethical imperialism [38,39]. If in a community, a team is told collective approval is sufficient or even preferable, some teams or professionals might feel uncomfortable and then insist on obtaining individual consent. At the same time, in the scenario outlined, the team member's discomfort indicates that reliance on the community leader's approval was more a matter of convenience than sensitivity to community decision-making norms and processes. This is not ideal. Not obtaining informed consent may be interpreted as a team failing to respect individuals' autonomy [1]. Some community members can even be further marginalized by a drone project if drone teams obtain informed consent from leaders who do not have the interests of the most vulnerable members of society as a priority. Ultimately though, this may be resolvable through clear and diplomatic communication. Drone teams could respect local cultures by obtaining local leaders' consent while also explaining the importance of abiding by guidelines that require drone teams to consult and obtain consent from the wider community. Perhaps this would be challenging, but perhaps not. These challenges can only be identified when Drones for Health teams start to work with local leaders to obtain consent that respects the cultural values and individuals of the community.

Participants in this study highlighted a need for better preparedness for situations where normal project consent and information processes are disrupted by unanticipated events. Though teams need to respect local culture and gain approval from community representatives, it does not mean that teams can completely disregard the need to hold community consultation or engagement activities. In addition, there are inconsistent consent practices in terms of where (urban vs. rural) and when (obtaining an individual's sample or using an individual's private space) teams seek individuals' consent to participate in the health project.

Regardless of whether a Drones for Health project is categorized as a public health initiative or a research project, it is imperative for teams to hold community consultation or engagement initiatives. Community consultation and engagement initiatives attempt to protect and respect an individual's autonomy and decision-making capacity both in the context of a public health initiative and collective consent [5,7,31]. Disclosing information about the public health initiative is crucial in protecting individuals' autonomy as it may provide them an opportunity to opt out or seek alternative care if possible [5]. Community engagement can potentially improve individuals' compliance with health intervention, increase individuals' trust, and allow individuals to prepare and take steps to minimize possible breaches of confidentiality [5,46]. Democratic, transparent decision-making procedures can help balance the interest of individuals and communities [2]. In fact, in certain cases, it has been recognized that community engagement or community consultation can act in lieu of an individual's informed consent [5,7]. The most appropriate methods must be determined endogenously, in consultation with communities. Informed consent processes must not be discarded in favor of community engagement and consultation because the latter is regarded as easier or faster. Ideally, the use of community engagement processes provides some opportunity for communities to be involved during the decision-making process of the public health intervention, including identifying what methods of consultation (individual or collective) are most likely to be successful in upholding norms of voluntary decision-making [5,7].

Not only did this study help us identify the context-specific concerns, challenges, and complexities of using Drones for Health, but in this case, it also allowed us to better understand and analyze the nuanced ethical and practical questions and challenges that may arise as teams set out to "obtain consent" for a project. We hope that this analysis will expand conversations on whether or not, from whom, and with attention to which diverse and context-specific considerations teams working to initiate drone projects in new locales engage in informed consent-seeking practices.

*4.1. Recommendations for Government Institutions*

(1) Given that people leading similar Drones for Health projects categorize them differently (research project versus public health initiatives), there appears a need for more discussion as to how these drone technology projects should be categorized to better determine teams' responsibilities to engage in individual informed consent processes. A document could be produced that explicitly outlines the activities.

(2) If projects are research-based and informed consent is needed, a protocol should be developed by drone teams in collaboration with stakeholders, such as government officials, local research ethics boards, non-governmental organizations, and traditional leaders, to support informed consent processes and ensure that consent tools are appropriate for the literacy level of the target population.

(3) Require drone teams to identify who is impacted by the Drones for Health project and develop appropriate consent plans for the affected groups.

*4.2. Recommendations for Drones for Health Teams*

(1) If the Drones for Health projects do not address urgent needs (i.e., flood), drone teams could visit the community more than once—once to perform the community consultation and the next time to perform the drone flights. Separating the tasks will give drone teams more time to focus on ensuring the community understands what is going on and not be rushed to perform the drone flights.

(2) Teams could work collaboratively with community gatekeepers to identify an appropriate and convenient time when most, if not all, community members are available to attend gatherings and drone demonstrations. The team should let community members know well ahead of time about the gatherings and demonstrations and the importance of attending them, so that community members can arrange to participate, while paying attention to local power dynamics and literacy levels.

**Author Contributions:** Conceptualization, V.J., E.N., L.D. and P.M.; methodology, V.J., E.N., L.D. and P.M.; interviews, V.J. and E.N.; formal analysis, V.J., E.N. and L.D.; writing—original draft preparation, V.J.; writing—review and editing, V.J., E.N., L.D. and P.M.; supervision, E.N. and L.D.; project administration, V.J. and E.N. All authors have read and agreed to the published version of the manuscript.

**Funding:** This research was funded by the Canadian Institute for Health Research, grant number 390089 (PI E. Nouvet).

**Institutional Review Board Statement:** This study was approved by the ethical committee of Western University (113823 Approved 13 May 2019).

**Informed Consent Statement:** Informed consent was obtained from all subjects involved in the study.

**Data Availability Statement:** Data available on request due to restrictions (e.g., privacy or ethical).

**Acknowledgments:** The authors would like to thank Matthew Hunt for his contributions to this paper. Additionally, the authors are grateful for Ishor Sharma, Ujash Sooriyakumaran, and Diego Sornoza's help with translating and transcribing the interviews. Additionally, the authors would like to also express their sincere gratitude to Gabriella Ailstock from VillageReach for sending the recruitment poster in the monthly UPDWG newsletter.

**Conflicts of Interest:** The authors declare no conflict of interest.

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
