# Peer review of "To Obtain Informed Consent or Not to Obtain Informed Consent? Drones for Health Programs in the Grey Zone between Research and Public Health"

_drones, doi:10.3390/drones7040247_

Round 1

Reviewer 1 Report

The reviewed article has a topical theme and a standard structure. From a methodological point of view, I recommend adding more factual information about the research conducted, its limitations, the number of participants and the researchers' own objectives and their fulfilment.

I recommend ending the first part of the introduction with the GAP found. Based on it, define the intent and goal of the paper. 

In the second part I recommend adding graphical objects - maps, tables, etc. to complete the factual aspect of the research. 

Attention in the text in the second part is used in the odesk Recruitment and sampling strategy, a different format.

In the third part I recommend to include appropriate tables of results that will better explain the actual results.

I recommend structuring the discussion and conclusion, e.g. - how the research plan was met, how the aim and objective of the paper was met, what other themes and challenges the research identified.

The Reccomendations section is very well done, I just recommend to group appropriately, e.g. - recommendations for civil society, for health institutions and for government institutions.

Author Response

Thank you for reviewing this paper and providing us with the opportunity to submit a revised draft of the manuscript “To obtain informed consent or not to obtain informed consent? Drones for health programs in the grey zone between research and public health.” for publication in Drones. We appreciate your time and effort.

Please find the edited manuscript attached with the tracked changes.  

And please find our response to your comments here: 

Reviewer’s Comment

Author Team’s Response

From a methodological point of view, I recommend adding more factual information about the research conducted, its limitations, the number of participants and the researchers' own objectives and their fulfilment.

Thank you for this feedback. We included additional information in lines 184-185 to address this. Information on participants could be found in 190-192 and 201-202.

I recommend ending the first part of the introduction with the GAP found. Based on it, define the intent and goal of the paper. 

The gap and goal of the paper can be found in lines 108-180.

In the second part I recommend adding graphical objects - maps, tables, etc. to complete the factual aspect of the research. 

Our team has considered this suggestion. We certainly appreciate tables and graphics in general. In this case, section 2 is very brief as we are directing readers to a previously published paper for a more detailed account of methods. Had we received this feedback on that paper, we would have included a map with it indicating the continents from which our interviews were sourced; but, we feel this would not be appropriate in the current paper as the methods are now published/represented in that prior publication. We hope the reviewer will understand.

Attention in the text in the second part is used in the odesk Recruitment and sampling strategy, a different format.

Thank you for catching this. The formatting has been changed to match the rest of the paper.

In the third part I recommend to include appropriate tables of results that will better explain the actual results.

A table has been added at the end of the section on line 504 to summarize the results.

I recommend structuring the discussion and conclusion, e.g. - how the research plan was met, how the aim and objective of the paper was met, what other themes and challenges the research identified.

Thank you for this feedback. We have added lines 565-571 to further clarify the project aim and the goals we achieved.

The Reccomendations section is very well done, I just recommend to group appropriately, e.g. - recommendations for civil society, for health institutions and for government institutions.

Based on this feedback, we have categorized the recommendations based on the different stakeholder groups.

Thanks,

Vyshnave 

Reviewer 2 Report

This paper is a timely and necessary addition to the drone literature - health projects are not always high on the agenda because they have less commercial value. I see no major problems with the article content and only have a few constructive comments that came to mind while reading.

In terms of the bigger picture, ethics and consent is a tricky area (a bit of a minefield) for drone health-related projects as the black and white can quickly merge to grey. For example health projects that conduct mapping by photographing large areas of land may obtain consent from a village elder but be infringing on the privacy and rights of multiple landowners, even if it is accidentally. Who is the priority in such a case for obtaining consent? What if one individual objects?

Literacy must be taken into account in the local setting - it may be difficult to ensure everybody truly understands the reason for and value of the project. On that note, dealing with conspiracy theories that might emerge after a project has begun may present issues for completing a project. Individuals may spread rumours that signing a consent form has tricked them into giving away rights. Guidelines for this scenario may be helpful for project teams.

Emphasis on sensitivities to local sociocultural and political conditions is just as important in project planning and should be implemented as part of the ethics approval process. This will avoid a one size fits all approach to project design.

I assume the next step is to suggest a set of (adaptable) guidelines for ethical drone use in health projects in consultation with appropriate health agencies etc.? I look forward to it!

Author Response

Thank you for reviewing this paper and providing us with such inisghtful comments that need to be considered when obtaining consent for drones for health projects. We appreciate your time and effort.

Please find the edited manuscript attached with the tracked changes.  

And please find our response to your comments here: 

Reviewer’s Comment

Author Team’s Response

In terms of the bigger picture, ethics and consent is a tricky area (a bit of a minefield) for drone health-related projects as the black and white can quickly merge to grey. For example health projects that conduct mapping by photographing large areas of land may obtain consent from a village elder but be infringing on the privacy and rights of multiple landowners, even if it is accidentally. Who is the priority in such a case for obtaining consent? What if one individual objects?

Thank you for these insightful comments. We agree with the trickiness of obtaining informed consent and like you, we hoped we were able to raise some of these complexities throughout the paper.

We briefly discussed this situation in lines 446-447 and then further clarified it in lines 452-454.

Literacy must be taken into account in the local setting - it may be difficult to ensure everybody truly understands the reason for and value of the project. On that note, dealing with conspiracy theories that might emerge after a project has begun may present issues for completing a project. Individuals may spread rumours that signing a consent form has tricked them into giving away rights. Guidelines for this scenario may be helpful for project teams.

We raised this question in lines 279-281 and then included a note about literacy levels in our recommendations in lines 582 and 607

Emphasis on sensitivities to local sociocultural and political conditions is just as important in project planning and should be implemented as part of the ethics approval process. This will avoid a one size fits all approach to project design.

We have added lines 439-441 to highlight to drone teams that they need to pay attention to the local sociocultural and political conditions when obtaining consent.

I assume the next step is to suggest a set of (adaptable) guidelines for ethical drone use in health projects in consultation with appropriate health agencies etc.? I look forward to it!

Thank you for all your insightful! We have been working with WeRobotics to support the development of their drone guidelines.

Thanks,

Vyshnave
